# Mitigating Weight Gain Side Effects by Reducing Sugar-Sweetened Beverage Consumption in Youth Newly Prescribed Second-Generation Antipsychotic Medication

**DOI:** 10.3390/nu18010024

**Published:** 2025-12-20

**Authors:** Kristin Bussell, Heidi Wehring, Susan dosReis, Raymond C. Love, Jason Schiffman, John Sorkin, Zhaoyong Feng, Sarah Edwards, Erin Hager, Elizabeth A. Dennis, Kathleen Connors, Kathryn McDonald, Meredith Roberts, Emily Wolfe, Shlomo Resnik, Gloria Reeves

**Affiliations:** 1Department of Family and Community Health, University of Maryland School of Nursing, Baltimore, MD 21201, USA; 2Department of Psychiatry, University of Maryland School of Medicine, Baltimore, MD 21201, USAkconnors@som.umaryland.edu (K.C.); meredith.roberts@som.umaryland.edu (M.R.); szresnik@gmail.com (S.R.); greeves@som.umaryland.edu (G.R.); 3Department of Practice, Sciences, and Health Outcomes Research, University of Maryland School of Pharmacy, Baltimore, MD 21201, USA; 4Department of Psychology, University of Maryland, Baltimore County, Baltimore, MD 21250, USA; 5Department of Psychology, University of California, Irvine, CA 92697, USA; 6Department of Practice Medicine, University of Maryland School of Medicine, Baltimore, MD 21201, USA; 7Department of Pediatrics, University of Maryland School of Medicine, Baltimore, MD 21201, USA; 8Department of Population, Family and Reproductive Health, Bloomberg School of Public Health, Johns Hopkins University, Baltimore, MD 21205, USA; 9Department of Family Medicine, University of Maryland School of Medicine, Baltimore, MD 21201, USA

**Keywords:** healthy lifestyle, sugar-sweetened beverages, water, weight gain, antipsychotic medication

## Abstract

**Background:** Antipsychotic medication (APM) can cause weight gain, insulin resistance, dyslipidemias, and an increased risk of developing type-2 diabetes and cardiovascular disease among youth. The study sought to increase water consumption, reduce sugar-sweetened beverage (SSB) intake, and prevent unhealthy weight gain via a healthy lifestyle intervention among youth newly started on a second-generation APM for psychiatric treatment. **Methods:** This randomized controlled trial enrolled 148 Medicaid-insured youth (ages 8–17) recently starting APM. The treatment group received both a biweekly home-delivery of bottled water and parental phone support from a family navigator. In-home visits conducted at baseline, three months, and six months assessed the participants’ height/weight and dietary intake. All participants received basic healthy lifestyle education emphasizing increased water intake and decreased SSB consumption. Longitudinal linear mixed models were conducted to examine between-group and within-group changes in BMI z-scores, and water/SSB intake. **Results:** No significant between-group differences in BMI z-score were found at three (*p* = 0.908) and six months (*p* = 0.919). However, the within-group increase in BMI z-score in the control group was significant from baseline to three months (*p* = 0.029). A between-group comparison found the treatment group significantly increased their water intake at three (*p* = 0.006) and six months (*p* = 0.002). No between-group differences were identified at three and six months for the reduction in SSB, although the treatment group did demonstrate a decrease from baseline to three months (*p* = 0.004). **Conclusions:** Neither group experienced unhealthy increases (>0.5%) in BMI z-score over the six months. Providing a safe/free water supply showed a superior improvement in water consumption in the treatment group, and an initial decrease in SSB. Further studies are needed to identify feasible, effective, and sustainable lifestyle interventions tailored to this at-risk population.

## 1. Introduction

The American Academy of Pediatrics (AAP) identified second-generation antipsychotic medication (APM) treatment as a significant risk factor for pediatric obesity [1]. These medications are approved by the Food and Drug Administration (FDA) for the treatment of pediatric bipolar disorder, autism, and schizophrenia. In community care, APMs are most commonly used to treat severe aggression and irritability [2]. Among APM-treated youth, approximately 60% experience weight gain side effects, and 30% have a clinically significant increase in BMI z-score > 0.5 [3]. Complex genetic, central nervous system, gut microbiome, and neuroendocrine underlying mechanisms have been implicated in the key APM obesogenic features of increased hunger and delayed satiety [4]. In the United States (US), prior authorization programs have been implemented to reduce unnecessary APM exposure [5], but poorly functioning youth may still require treatment.

Medication-induced obesity can negatively impact metabolic health, including introducing an increased risk of hypertension, type 2 diabetes, dyslipidemia, and nonalcoholic fatty liver disease [6]. Equally concerning, obesity-related stigma and bullying may lead to depression, low self-esteem, social isolation, and impaired school performance [7]. Youth with rapid weight gain may also become school-avoidant due to the impact on body image and self-esteem. Obesity-related side effects also may have a negative impact on medication adherence and engagement in care [8]. Thus, APM-induced obesity can broadly impair child functioning.

Although youth are at a greater risk of APM-induced obesity than adults [9], there are no universally accepted pediatric healthy lifestyle interventions to mitigate this side effect. The relevant practice guidelines on pediatric antipsychotic treatment from the American Academy of Child and Adolescent Psychiatry (AACAP) [10] are outdated. The recent AAP guidance on the treatment of pediatric obesity highlights Intensive Health Behavior and Lifestyle Treatment as the foundational approach to mitigating excessive weight gain in children and adolescents; however, this treatment requires a multidisciplinary team approach and minimum “dosing” of 26 contact hours with a multidisciplinary team over 3–12 months [1]. This treatment may not be accessible or practical for families with competing mental health appointment demands. There is a compelling need for feasible, evidence-based strategies to mitigate weight gain side effects.

Research studies on APM-induced weight gain management have primarily focused on pharmacologic interventions by switching the APM or adding a weight loss drug [11]. Pediatric weight gain is common at low doses and not consistently dose-dependent [10]; therefore, lowering the dose is often ineffective. Metformin has the most empirical pediatric evidence for the treatment of APM-induced weight gain [11], but it may not be appropriate for youth who are not yet overweight or obese. Switching APM early in care can destabilize the youth’s mental health. The research on behavioral interventions is quite limited. One pilot study [12] reported a modest decrease in adiposity and hepatic fat in overweight and obese youth receiving a pediatric evidence-based weight loss intervention (Traffic Light Plan). A larger study [13] reported no difference between an intensive healthy lifestyle intervention (food/activity logs, weight counseling, and pedometer use) compared to a low-intensity counseling session.

Rice and Ramtekkar [14] proposed practical healthy lifestyle targets to manage metabolic syndrome in APM-treated youth, which included reducing sedentary behavior, improving stress management, promoting healthy sleep, and improving healthy food and beverage choices. They specifically recommended that parents replace sugar-sweetened beverages (SSBs) with low-calorie drink options at home. SSBs are liquids with added sugar, such as calorically sweetened soda or fruit punch that is not 100% fruit juice. The World Health Organization identifies SSB consumption as a major contributor to pediatric obesity [15]. The low satiety of these highly palatable beverages results in excessive calorie consumption, and spikes in insulin and glucose occur after SSB intake [16]. APM-treated youth may be quite vulnerable to excessive SSB intake, since anti-cholinergic side effects increase thirst.

A systematic review of water promotion strategies to reduce SSB intake (e.g., water dispensers) concluded that water promotion alone is not sufficient to decrease SSB consumption and that health education is also recommended [17]. A 2023 systematic review of randomized trials lasting six months or longer focused on SSB replacement interventions reported an estimated 0.31 kg/m^2^ reduction in BMI equivalent to 0.5–1 kg in children and adults over the intervention period [18]. Home-based water delivery is a simple intervention that involves a low appointment burden, may have collateral benefits for other family members, and is appropriate for both obesity prevention and reduction. However, this type of intervention has not been rigorously studied for APM-treated youth.

This paper reports on a six-month randomized trial of health education versus a healthy lifestyle intervention that includes health education, a healthy lifestyle coach, and home delivery of bottled water. We report on changes in the BMI z-score, SSB intake, and water consumption at three and six months.

## 2. Materials and Methods

### 2.1. Ethics Statement

The Institutional Review Boards of the University of Maryland, Baltimore, and the Maryland Department of Health approved the study protocol. Informed written consent was obtained from all adult participants (parent/guardian), and assent from youth participants enrolled in the study. The study is registered with clinicaltrials.gov (NTC02877823).

### 2.2. Participants/Design

Eligible youth aged 8–16 years old and approved by a State Medicaid antipsychotic prior authorization program for antipsychotic treatment were offered enrollment in the study (see Figure 1). Parents received a letter by mail notifying them that their child may be eligible to participate in a voluntary study and were provided an addressed/stamped response envelope with options to decline or be contacted by phone to receive more information. Exclusion criteria included any of the following: youth who were non-English speaking, wheelchair-bound (unable to use a pedometer to track activity), residing in a residential facility or foster care, and non-verbal, or have an IQ < 55 (measured using the Wechsler Abbreviated Scale of Intelligence matrix reasoning subtest). Participants were not excluded based on diagnosis or specific APM treatment. Parent participants were either a biological parent or a legal guardian who spoke English. In total, 148 youth–parent/guardian dyads were enrolled.

In this longitudinal randomized control trial, youth were randomized to the control group or the treatment group using a binary method, stratifying for age (<12, ≥12), sex (biological male/female), and BMI percentile category (<85th %, ≥85th %). Stratification on these factors was due to differences in typical growth patterns in these sub-groups [19], as well as concerns that weight change may differ in those of normal weight and those who were already overweight or obese (i.e., obesity prevention versus obesity intervention). Study data were collected at baseline, 3-, and 6-month time points.

### 2.3. Interventions

Healthy lifestyle education (HLE) was provided to all parent participants in both the control and treatment groups during in-home study visits at baseline, 3, and 6 months by a research study team member. The HLE provided was based on the American Academy of Pediatrics Healthy Lifestyle Healthiest Weight Guidelines and included the following information: increase intake of water (≥48 oz daily) and fruits/vegetables (≥5 daily), decrease intake of sugar-sweetened beverages and foods high in sugar/fat, limit fast food and/or restaurant food to once per week, reach 1 h per day of physical activity, and limit screen time to maximum of 2 h per day. Parents were taught how to read food labels to distinguish sugary beverages from non-caloric beverages, 100% fruit juice, and unflavored milk.

Treatment group: This group received HLE at study visits plus biweekly home delivery of bottled water for the household (up to 6 people) based on daily consumption of 48 oz by the youth and 24 oz for each household member. This protocol was based on a study [20], which provided home delivery of non-caloric beverages to overweight and obese youth and allocated supplies to household members to avoid competition over available beverages. However, we exclusively delivered water instead of a mix of water and “diet beverages” to optimize the health benefits of water hydration. Youth were provided a medical note allowing them to bring and consume their bottled water at school. A family navigator (trained parent peer support specialist) contacted the parent weekly by phone the first month, and then every two weeks to offer emotional support and coaching on lifestyle changes (e.g., praise the child for incremental change rather than a punitive approach). The family navigator model we developed [21] also allowed for flexible communication by offering additional ad hoc telephone calls as needed by the parent, including on weekends and evenings. Family navigators were trained by the study team on motivational interviewing techniques, healthy lifestyle education, and protocols to manage any unexpected psychiatric safety concerns reported by the parent (an on-call licensed child mental health clinician was available for any safety concerns after hours and weekends). The role of the FN was to provide individualized emotional support and to address common barriers to change (e.g., use of sugary beverages to reward behavior). Participants were assigned a consistent family navigator to work with during the study. Additionally, treatment-group youth participants were provided a pedometer, and parents were asked to monitor/track their child’s steps and encourage activity. The FN emphasized incremental change in activity (e.g., reducing sedentary activity) and focusing on preferred child and family activities (e.g., video game dance activities) rather than regimented exercise, consistent with a “small steps for change” approach [22]. This approach minimized parent–youth conflict, which could be triggered by expecting major changes, especially while the child was still experiencing psychiatric instability.

### 2.4. Measures

Youth dietary intake was measured at baseline, 3 months, and 6 months with the online Automated Self-Administered 24-h (ASA-24) dietary assessment tool (version 2020). This computerized program, developed by the National Institute of Cancer (Bethesda, MD, USA), is based on the gold standard assessment method of the United States Department of Agriculture (USDA) Automated Multiple Pass Method (AMPM) 24 h dietary recall [23].

The assessment was administered by study team members, rather than self-administered by the participant, to ensure complete data collection. Specific study team members received training, observation, and oversight in congruence with the USDA AMPM procedures and methods to ensure fidelity. At each study time point (baseline, 3 months, and 6 months), 24 h dietary recalls were obtained three times, including two weekdays and one weekend day, and administered in accordance with the National Health and Nutrition Examination Survey (NHANES) standardized procedures [24,25]. In-person assessments were conducted at the study visit (baseline, 3 months, and 6 months), followed by 2 additional assessments by phone. All children participated in the assessments, and parents were present to provide details related to food content and preparation.

The ASA-24 program provides a comprehensive dietary analysis that includes a total of 65 nutrients and 37 food groups [26]. Foods and beverages reported via ASA-24 are automatically coded using food codes from the USDA’s Food and Nutrient Database for Dietary Studies (FNDDS 2015-16). The FNDDS food codes for beverages, including beverages with additions, were sorted into mutually exclusive beverage categories adapted from the What We Eat in America (WWEIA) Food Categories [27]. The water category included the FNDDS codes under the WWEIA Food Category for plain water, including tap water and bottled water, and unsweetened carbonated water. The sugar-sweetened beverage category included the following FNDDS codes under the WWEIA Food Category: sweetened beverages, soft drinks, fruit drinks, sport and energy drinks, calorically sweetened coffee, calorically sweetened tea, and calorically sweetened water. Grams and calories from beverage additions were added to the corresponding beverages. Daily total grams and calories from each category were calculated by summing the beverages reported in each category. Beverages added to foods were excluded from the beverage categories (e.g., milk added to cereal). All beverages were reported in grams and converted to fluid ounces using the conversion factor, 30 g = 1 fluid oz, and then assigned to the corresponding beverage category (e.g., participant reported drinking tea and added their own sugar).

Height and weight of the youth participants were measured at each study visit utilizing a portable stadiometer and the Tanita TBF-300 scale with standardized procedures we used in our prior research [28]. Child BMI z-score was calculated using the Baylor School of Medicine Children’s Nutrition Research Center BMI percentile for age calculator [29]. At each in-home study visit (baseline, 3 months, and 6 months), the parent was asked to show the research team member the APM prescription bottle to confirm the child had a medication supply at home. All prescribed and over-the-counter medications, as well as supplements, were recorded. Parents were encouraged to contact their child’s prescriber if they had any concerns about medications. If they did not appear to have a supply of the APM medication at home, they were advised to contact the child’s provider to confirm the medication regimen plan. Parents were informed that a clinician member of the research team could assist with referrals for their child if discharged from a practice or the family relocated to another part of the State, but the study team did not provide any psychiatric medication treatment or care.

Pandemic-related protocols: Participant enrollment started in February 2017. All in-person study visits were suspended at the end of March 2020 due to pandemic research restrictions and, therefore, did not meet our enrollment goal of 180 participant dyads.

### 2.5. Statistical Analysis

In calculating our power analysis, we assumed that over a short time interval (3 to 6 months), BMI percentile z-scores would be strongly correlated within participants (r = 0.6 to 0.7). Based on prior experience with data requests for Medicaid preauthorization antipsychotic prescriptions and with our previous study using this enrollment strategy, we estimated it to be feasible to enroll an average of 4 to 5 new participants per month over 36 months. Assuming 20% of participants would withdraw from the study before 6 months, this would leave between 57 and 72 participants per group. Under these assumptions, for *n* = 57 (4 participants per month), we would have power > 0.80 to detect an effect size of d = 0.38 s.d. if r = 0.7, or d = 0.43 s.d. for r = 0.6. 

Youth and parent baseline characteristics were compared between the control and treatment groups using Pearson’s Chi-Square or Fisher’s Exact test for categorical data, and Student’s *t* tests or Wilcoxon Rank Sum tests for continuous data. Youth BMI z-scores at different time points (baseline, 3-month, and 6-month) were examined in mean, standard deviation, minimum, lower quartile, median, upper quartile, and maximum. Spaghetti plots were generated to visualize the direction and magnitude of change in BMI z-scores in each study group with model-based means and 95% confidence intervals for each visit. Linear mixed-effects models (SAS procedures mixed with a repeated statement) were used to evaluate the relationship between the intervention and longitudinal changes in youth BMI z-scores. In the first model, the dependent variable was the child’s BMI z-score. Independent variables included time point, intervention, and an interaction of intervention and time point. The second model was adjusted by adding the child’s race and ethnicity to the first model. AIC (Akaike Information Criteria) were used to select the covariance structure (unstructured, compound symmetry, first-order regression, and variance components) that best accounted for the serial autocorrelation or repeated measures from the same youth. Model-based estimates of mean BMI z-score at each time point and treatment were compared.

These same procedures were performed to analyze changes in water consumption and sugar-sweetened beverage intake within each study group and between the treatment and control groups at each time point. All *p*-values were computed for 2-tailed tests, and the level of significance was set at 0.05. All statistical analyses were performed using SAS version 9.4 [30].

## 3. Results

The baseline characteristics of the parents and youth revealed no statistically significant differences between the control and treatment groups. The sample consisted of a relatively even number of males and females, with a mean age of 13 years old, and was primarily composed of either white (47%) or black (40%) individuals (Table 1). Parents were primarily mothers in their forties, white (54%) or black (41%), with household incomes under $40,000 (Table 2).

### 3.1. Youth Medication Treatment

All enrolled youth participants were treated with APM antipsychotics. The three most common APMs prescribed in both groups were aripiprazole (43.6% treatment; 39% control), risperidone (33.3% treatment; 31.0% control), and quetiapine (treatment 9.0%, control 11.4%). One child participant in each group was treated with two APMs concurrently at the time of enrollment. At baseline, 39.7% of the treatment group and 38.6% of the control group were co-prescribed stimulant medication for the treatment of Attention Deficit Hyperactivity Disorder. No parents of the youth in our sample reported their child was receiving any weight-loss dietary supplements during their study participation.

### 3.2. Participants Who Completed All Three Visits

Among study completers, inconsistent APM treatment over the six months was common. Only 48.29% of the treatment group and 55.8% of the control group were (1) prescribed the same APM over six months and (2) able to demonstrate any home supply of the APM at each visit, confirmed by a visual inspection of the child’s prescription container. Approximately 8.9% of parents with a child enrolled in the treatment group and 9.3% of parents with a child enrolled in the control group reported their child was not receiving any APM treatment at both the 3- and 6-month study visits. Moreover, among study completers, 12.5% of the treatment group and 11.6% of the control group experienced a switch to a different APM over the course of the study.

### 3.3. Change in BMI Z-Score

At baseline, no significant differences were found in youth BMI z-scores (Table 3) between study groups, which persisted over the course of the 6-month study (Figure 2). From baseline to six months, the mean BMI z-score estimated by linear mixed-effects models increased from 1.11 to 1.19 in the control group and from 1.13 to 1.21 in the treatment group (mean difference, treatment vs. control, −0.0089; *p* = 0.909), with no significant difference in change between the groups (Table 4). The differences in the model adjusted for race and ethnicity were also nonsignificant. However, a statistically significant increase in BMI z-score from baseline to three months was found within the control group (*p* = 0.029) (Table 5), which remained significant (*p* = 0.032) in the model adjusted for race and ethnicity.

### 3.4. Changes in Beverage Consumption

#### Sugar-Sweetened Beverage Intake

At baseline, the sugar-sweetened beverage (SSB) intake was very similar in both groups, and, although both groups had a decrease in SSB intake, there was no significant difference between groups (Table 3). From baseline to six months, the mean SSB consumption estimated by linear mixed-effects models decreased from 10.82 to 9.76 oz in the control group and from 10.30 to 7.45 in the treatment group (Figure 3) (mean difference, treatment vs. control, −2.09; *p* = 0.409), with no significant difference in improvement (Table 4). The differences in the model adjusted for age, race, gender, and ethnicity were also nonsignificant. We did detect a significant decrease in the mean SSB consumption at three months (−3.97, *p* = 0.004) relative to baseline in the treatment group (Table 5).

### 3.5. Water Intake

At baseline, there were no significant differences in water intake between the control and treatment groups but the mean water intake for the treatment group increased significantly compared to the control group from baseline to three months (*p* = 0.006), and, although the treatment group intake decreased slightly from three to six months, it remained statistically significant (*p* = 0.002) compared to the control group (Table 3). From baseline to six months (Figure 4), the mean water intake estimated by linear mixed-effects models increased from 23.55 to 36.78 oz in the treatment group and decreased from 22.29 to 20.84 oz in the control group (Figure 4) (mean difference, treatment vs. control, 14.67, *p*-value = 0.0009), with a significant difference in improvement (Table 4 and Table 5), and this remained significant after adjusting for age, gender, race, and ethnicity at baseline (mean difference, treatment vs. control, 14.65, *p*-value = 0.0009).

## 4. Discussion

The main findings of this study were that the healthy lifestyle intervention did not result in a superior reduction in BMI z-score or SSB intake compared to the control condition, but it did result in a greater increase in youth water consumption. Both groups, despite recently initiating obesogenic APMs, only had negligible increases in the BMI z-score, and there was no significant difference between them. It is possible that the lack of differences between groups was because the health education provided to both groups was effective in mitigating APM-induced weight gain. The study included a family-based approach and intervention duration of six months, which have been identified as important study design features in effective pediatric obesity interventions [31]. Our approach also avoided common barriers to healthy lifestyle changes identified by parents of children with special mental health needs. Commonly reported barriers are the use of overly restrictive or stressful education goals that may trigger child emotional dysregulation, the high time burden for healthy lifestyle changes, the need for access to specialized nutrition and exercise experts, and the lack of consideration as to the medication’s side effects in educational interventions [32]. In our past research, a low-intensity healthy lifestyle education intervention provided to parents of obese and overweight APM-treated youth resulted in a statistically significant reduction in child overall daily calories (*p* = 0.002) and sugar intake (*p* = 0.008) [28].

The lack of significant weight gain from baseline to six months may also have been due to the early discontinuation of the baseline APM in a substantial portion of participants. Among study completers in each group, approximately one out of five participants either switched APM or discontinued any APM treatment during the course of the study. The early discontinuation of APM treatment is not uncommon in community care. In a Finnish register-based national study, approximately 35% of APM-treated youth under 18 years old received less than 50 days of treatment [33]. A naturalistic follow-up study of pediatric first episode psychosis patients who started APM treatment during a hospital admission reports 45% of youths discontinued APM treatment by six months [34]. The discontinuation of APM may be due to insufficient efficacy, side effects, parent/youth preferences, or a combination of these reasons. In this study, it is possible that participants who discontinued APM early were the youths who experienced the most weight gain.

The intervention group was not superior to the control group in the reduction in SSB consumption. Both groups demonstrated a modest reduction in SSB intake, and the intervention group had a significant decrease from baseline to three months. Reducing SSB intake may be challenging for parents who have a child with special mental health needs, since parents often use SSBs to reward behavior, referred to as “instrumental feeding” [35]. Moreover, SSB intake may occur in community settings, such as sugar-flavored milk provided with school lunches. Families who have a high appointment burden may rely on fast food meals, which often “bundle” inexpensive kid meals with limited beverage options or serve beverages in unlabeled containers (e.g., fountain sodas) [36]. Youth who are obese report a higher consumption of food outside of the home compared to non-obese youth [37], and unhealthy beverage options (e.g., juice boxes with added sugar) may also be less expensive than bottled water.

Water consumption improved significantly among children who received home water delivery. An expert consensus group, composed of representatives from major US nutrition and health national organizations, recently disseminated guidance on pediatric daily water intake needs [38]. The consensus group strongly emphasized that “water should be the primary beverage” that youth consume. In our study, the baseline mean daily water intake for the control (22.1 oz) and treatment (23.7 oz) groups was well below the recommended range of water consumption for older adolescents (29–88 oz) and on the low end of the recommended intake for youth aged 9–13 years old (22–61 oz). It is possible that water intake before drug initiation was even lower, since APM may stimulate thirst due to the anticholinergic side effect of decreased salivation [39]. Water intake only increased in the intervention group, indicating that education alone was not sufficient to increase water consumption.

Adequate water intake may be especially beneficial for youth with special mental health needs due to the potential influence on mood and cognition. Water is fundamental in maintaining the osmotic pressure of cerebrospinal fluid, plasma volume, and cerebral blood flow, and even mild dehydration (1–2% loss of body mass) may negatively affect cognitive function and emotional regulation [40]. Youth are generally more prone to dehydration than adults since youth tend to be more physically active, have a greater surface to mass area, and their water intake is more likely to be restricted or require permission (e.g., school time) [41]. Hydration status has been reported to impact memory formation, comprehension, and perception. In one study of prepubertal youth, a higher water intake was associated with better response inhibition functioning on a cognitive task [42]. A review of hydration interventions on cognitive performance suggested that the acute consumption of approximately 250 mL of water 20–60 min before challenging academic engagement may improve children’s performance on cognitive tasks [43]. Data from a large national study of Korean youth reported that lower plain water consumption was significantly associated with higher rates of self-reported depression [44], although it is unclear if interventions to increase youth water consumption improve mood.

Tap water consumption could improve hydration status without the added cost of bottled water delivery. However, there are significant youth and parent barriers to increasing youth tap water intake. A review of studies of adult perceptions of US tap water safety reported a decline in the perceived safety of tap water over the past five years [45]. This trend may impact parents’ willingness to model tap water consumption as well as allow their child to drink tap water at home. In our study, parents were overwhelmingly positive about receiving water shipments, as they frequently expressed concerns about the safety of tap water at home or at their child’s school. Even when safe water is available, youth may deliberately avoid hydration in school settings due to discomfort using school restrooms. This aversion may be due to fears of school bullying in an unsupervised setting or concern that the restroom may not be clean after frequent use [46]. Research is needed on effective strategies to increase tap water consumption in families who have a child with special mental health needs.

### Limitations

The study had several limitations. The pandemic impacted our ability to conduct home visits and complete the home delivery of water (supply chain issues), resulting in the early termination of the study, and falling short of the a priori target recruitment numbers. Approximately 97% of participants completed a baseline visit before the pandemic shutdown, but the study retention rate (67%) was lower than anticipated (80%) due to the early termination of participants who were actively enrolled during the pandemic shutdown. Another limitation of the study was the use of self-reporting to assess SSB and water intake. Youth may have had difficulty recalling beverages consumed outside of the home, and they may have minimized SSB consumption or enhanced water intake due to social desirability bias. Moreover, we included all APM-treated youth, introducing significant heterogeneity in terms of clinical diagnoses and medications prescribed. Most of the sample in each group were prescribed aripiprazole, quetiapine, or risperidone, which are the three APMs most commonly prescribed for youth in the US and Europe [47]. A recent review reported these medications have an intermediate-risk weight gain category, less than olanzapine and clozapine, and more than ziprasidone [48], which differs from adult studies, in which aripiprazole is a much lower-risk agent [10].

Another limitation is that we did not report on APM dosing. The medication prior authorization program approves APM treatment and restricts the start and maintenance dose ranges according to evidence-based practice, thereby reducing the likelihood of outlier dosing. Co-prescribed stimulant medication may have had some impact on weight, but the stimulant treatment was similar in both groups. Stimulant treatment for co-morbid attention deficit hyperactivity disorder is common among obese APM-treated youth in community care, and it is unclear how protective these medications may be against weight gain. The diagnostic heterogeneity of the sample did not allow us to personalize the intervention for specific sub-groups of youth who may have greater difficulty with healthy lifestyle changes. For example, youth with more impulsive symptoms may have greater difficulty making healthy beverage choices. Finally, newer pediatric APM medications have become available since the study was first designed (e.g., lurasidone), which may have a lower obesity risk than the drugs prescribed in this study.

A strength of the study was the home-based approach, which may be more feasible for busy families with a high appointment burden. The recent availability of remote monitoring equipment, which allows the direct transmission of weight data from the home to clinicians, can also support options to conduct this type of study virtually. Moreover, the novel recruitment strategy allowed us to identify a state-wide population of youth who might benefit from a side effect intervention program. Most states have APM prior authorization programs, and this approach could be replicated in other US settings.

## 5. Conclusions

The home-delivered bottled water did not result in the mitigation of antipsychotic-induced weight gain, although it is notable that neither group experienced significant weight gain over the six-month study. Several other positive findings are promising for further testing in future studies. Most notable is the statistically significant increase in water consumption within the treatment group at both the three- and six-month time points, indicating access to a safe water supply may result in improved water consumption. Additionally, increasing water consumption may result in a decreased intake of SSB as demonstrated by the treatment groups’ decrease in SSB intake from baseline to three months (*p* = 0.004), and the continued decrease from three to six months. Another aspect to examine is the impact of healthy lifestyle education, which both groups received, on diet and physical activity, and the potential effects on weight changes. Future directions include an analysis of the dietary and activity data from this study and examining the relationship between changes in nutrition and activity levels and changes in weight status. Additionally, a detailed examination of the dietary intake of participants may shed light on specific patterns associated with antipsychotic treatment. A qualitative inquiry should be included in future study designs to identify specific strategies that fit the special needs of this unique population of youth with mental illness. Key elements are the feasibility and sustainability of lifestyle behaviors, given that mental illness is often chronic, requiring ongoing psychopharmacological treatment. Mental illness occurs in people of all socioeconomic levels, so studies should expand recruitment beyond only the Medicaid-insured population to examine the differences and determine generalizable findings.

## Figures and Tables

**Figure 1 nutrients-18-00024-f001:**
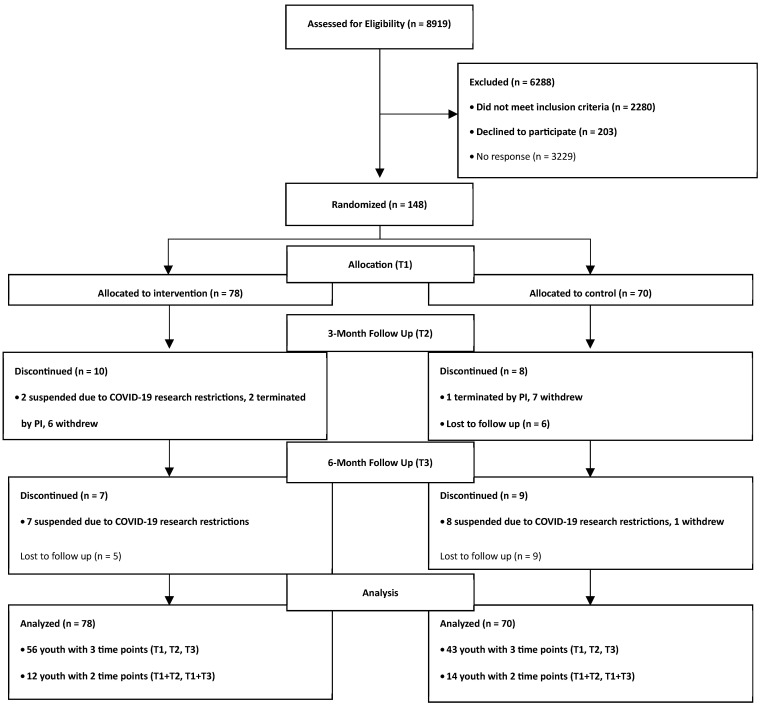
Enrollment consort diagram.

**Figure 2 nutrients-18-00024-f002:**
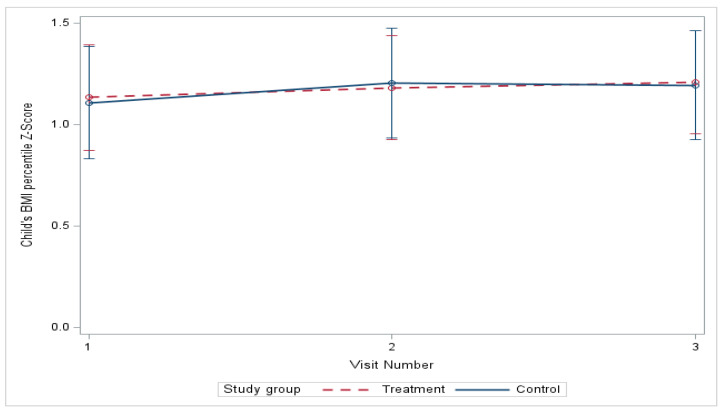
Change in BMI percentile z-score over 6 months.

**Figure 3 nutrients-18-00024-f003:**
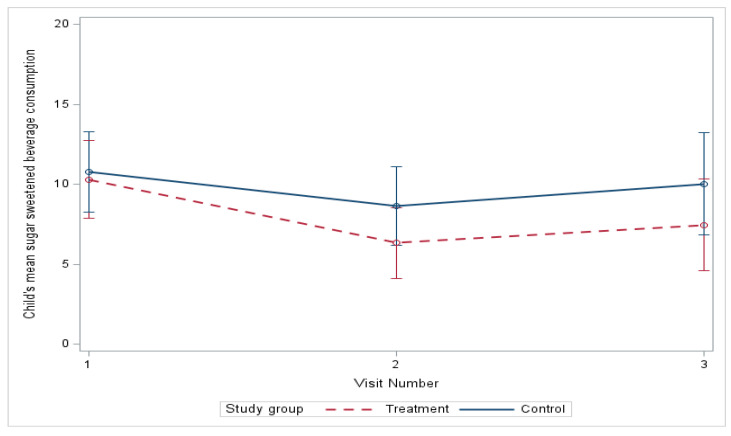
Change in sugar beverage consumption over 6 months.

**Figure 4 nutrients-18-00024-f004:**
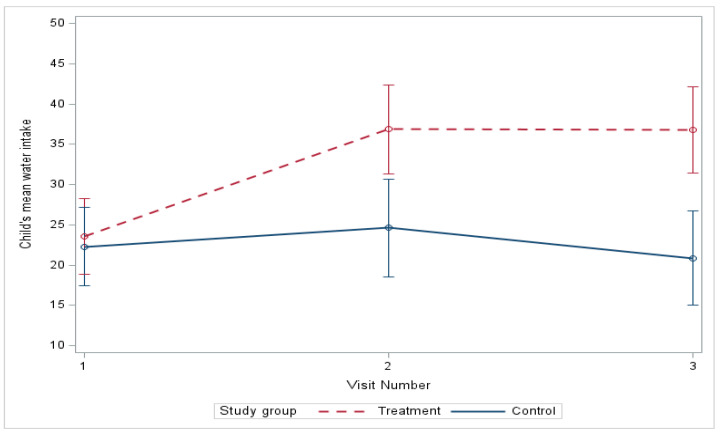
Change in water intake over 6 months.

**Table 1 nutrients-18-00024-t001:** Characteristics of youth at baseline.

Variables	All (N = 148)	Control (N = 70, 47%)	Treatment (N = 78, 53%)	*p*-Value
Age, years				
Mean (SD) *	13 (2)	13 (2)	12 (3)	0.842
Sex (*n*/%)				
Male	66 (44.6)	32 (45.7)	34 (43.6)	0.795
Female	82 (55.4)	38 (54.3)	44 (56.4)	
Race (n/%)				
White	70 (47.3)	35 (50.0)	35 (44.9)	0.798
Black	59 (39.9)	26 (37.1)	33 (42.3)	
Other race **	19 (12.8)	9 (12.9)	10 (12.8)	
Ethnicity (*n*/%)				
Hispanic or Latino				
NO	131 (88.5)	63 (90.0)	68 (87.2)	0.591
YES	17(11.5)	7(10.0)	10 (12.8)	
Primary Diagnosis (*n*/%)				
Disruptive Mood Dysregulation	24 (16.4)	9 (13.0)	15 (19.5)	0.612
Bipolar Disorder	19 (13.0)	11 (15.9)	8 (10.4)	
Mood Disorder NOS	9 (6.2)	4 (5.8)	5 (6.5)	
Major Depressive Disorder	32 (21.9)	12 (17.4)	20 (26.0)	
Schizophrenia	7 (4.8)	5 (7.2)	2 (2.6)	
Autism Spectrum Disorder	25 (17.1)	13 (18.8)	12 (15.6)	
Attention Deficit Hyperactivity	22 (15.1)	10 (14.5)	12 (15.6)	
Intellectual Disability	1 (0.7)	1 (1.4)	0	
Generalized Anxiety Disorder	3 (2.0)	1 (1.4)	2 (2.6)	
Post-Traumatic Stress Disorder	4 (2.7)	3 (4.4)	1 (1.3)	
N missing	2			
Repeated a grade in school (*n*/%)				
No	126 (85.1)	60 (85.7)	66 (84.6)	0.851
Yes	22 (14.9)	10 (14.3)	12 (15.4)	

* N, number of participants; SD, standard deviation; ** other race includes Asian, Hawaiian Pacific, and multi-race.

**Table 2 nutrients-18-00024-t002:** Characteristics of parents at baseline.

Variables	All (N = 148)	Control (N = 70)	Treatment (N = 78)	*p*-Value
Age in years (continuous)				
Mean (SD)	43 (9)	44 (10)	42 (9)	0.183
Sex (*n*/%)				
Male	14 (9.5)	5 (7.1)	9 (11.5)	0.362
Female	134 (90.5)	65 (92.9)	69 (88.5)	
Race (original) (*n*/%)				
White	80 (54.0)	40 (57.1)	40 (51.3)	0.931
Black	60 (40.5)	27 (38.6)	33 (42.3)	
Asian	4 (2.7)	2 (2.9)	2 (2.6)	
Hawaiian Pacific	1 (0.7)	0	1 (1.3)	
>1 race	3 (2.0)	1 (1.4)	2 (2.6)	
Race (new) (*n*/%)				
White	80 (54.0)	40 (57.1)	40 (51.3)	0.764
Black	60 (40.5)	27 (38.6)	33 (42.3)	
Other race	8 (5.4)	3 (4.3)	5 (6.4)	
Hispanic/Latino Ethnicity (*n*/%)				
NO	134 (91.2)	62 (89.9)	72 (92.3)	0.601
YES	13 (8.8)	7 (10.1)	6 (7.7)	
N missing	1			
Household Income (*n*/%)				
<20,000	49 (33.3)	23 (33.3)	26 (33.3)	0.836
20,000–40,000	51 (34.7)	26 (37.7)	25 (32.0)	
40,000–60,000	23 (15.6)	10 (14.5)	13 (16.7)	
60,000–80,000	12 (8.2)	5 (7.2)	7 (9.0)	
80,000–100,000	7 (4.8)	4 (5.8)	3 (3.8)	
>100,000	5 (3.4)	1 (1.4)	4 (5.1)	
N missing	1			
Employment Status				
Employed	66 (44.6)	31 (44.3)	35 (44.8)	0.812
Unemployed	72(48.6)	33(47.2)	39(50)	
Student	4(2.7)	2(2.8)	2(2.6)	
Retired	6(4.1)	4(5.7)	2(2.26)	

**Table 3 nutrients-18-00024-t003:** Distribution of BMI z-score, sugar-sweetened beverage, and water by time point.

Study Visit Time Point	Treatment		Control		*p*-Value
BMI Z-Score	N	M (SD)	N	M (SD)	
Visit 1	77	1.13 (1.15)	70	1.11 (1.17)	0.901
Visit 2	57	1.11 (1.08)	43	1.22 (0.97)	0.802
Visit 3	43	1.29 (1.04)	34	1.04 (1.04)	0.257
**Sugar-Sweetened Beverage** **Intake (ounces)**					
Visit 1	72	10.34 (8.74)	67	10.82 (11.96)	0.532
Visit 2	64	6.33 (8.02)	52	8.38 (10.01)	0.449
Visit 3	52	7.42 (10.74)	42	9.76 (10.34)	0.089
**Water Intake (ounces)**					
Visit 1	72	23.70 (18.28)	67	22.11 (21.84	0.641
Visit 2	64	37.14 (23.48)	52	25.18 (22.32)	0.006
Visit 3	52	36.35 (20)	42	20.53 (18.57)	0.0002

M, mean; N, number of participants; SD, standard deviation.

**Table 4 nutrients-18-00024-t004:** Longitudinal Linear Mixed Model 1 comparison of intervention effect on outcome variables (control vs. treatment) between each time point.

Difference in Effect Between Treatment and Control	Estimate	Standard Error	*p*-Value
**Difference in BMI Z-Score**			
Visit 3 vs. Visit 1	−0.00894	0.07838	0.909
Visit 2 vs. Visit 1	−0.04653	0.05745	0.419
Visit 3 vs. Visit 2	0.03759	0.05227	0.473
**Difference in Sugar-Sweetened Beverage Intake**			
Visit 3 vs. Visit 1	−2.0928	2.5257	0.409
Visit 2 vs. Visit 1	−1.8417	1.9841	0.355
Visit 3 vs. Visit 2	−0.2511	2.5379	0.921
**Difference in Water Intake**			
Visit 3 vs. Visit 1	14.6695	4.3106	0.0009
Visit 2 vs. Visit 1	10.9708	3.6309	0.003
Visit 3 vs. Visit 2	3.6987	4.2471	0.385

**Table 5 nutrients-18-00024-t005:** Longitudinal Linear Mixed Model 1 within and between-group comparisons of mean BMI z-score, sugar-sweetened beverage, and water by treatment group and time point.

Comparison	Estimate	Standard Error	*p*-Value
**BMI Z-Score**			
**Treatment vs. Control**			
Visit 1	0.02466	0.192	0.898
Visit 2	−0.02187	0.1886	0.908
Visit 3	0.01572	0.1864	0.933
**Control**			
Visit 3 vs. Visit 1	0.08503	0.05874	0.149
Visit 2 vs. Visit 1	0.09536	0.04330	0.029
Visit 3 vs. Visit 2	−0.01033	0.03899	0.791
**Treatment**			
Visit 3 vs. Visit 1	0.07609	0.05189	0.198
Visit 2 vs. Visit 1	0.04884	0.03776	0.198
Visit 3 vs. Visit 2	0.02725	0.03481	0.435
**Sugar-Sweetened Beverage**			
**Treatment vs. Control**			
Visit 1	−0.4615	1.7657	0.794
Visit 2	−2.3032	1.664	0.169
Visit 3	−2.5544	2.1761	0.243
**Control**			
Visit 3 vs. Visit 1	−0.7597	1.8601	0.684
Visit 2 vs. Visit 1	−2.1273	1.4573	0.147
Visit 3 vs. Visit 2	1.3677	1.888	0.470
**Treatment**			
Visit 3 vs. Visit 1	−2.8525	1.7085	0.097
Visit 2 vs. Visit 1	−3.969	1.3464	0.004
Visit 3 vs. Visit 2	1.1165	1.6961	0.511
**Water**			
**Treatment vs. Control**			
Visit 1	1.2637	3.3986	0.711
Visit 2	12.2345	4.1436	0.004
Visit 3	15.9333	3.9956	0.0001
**Control**			
Visit 3 vs. Visit 1	−1.4447	3.1792	0.65
Visit 2 vs. Visit 1	2.3197	2.6887	0.39
Visit 3 vs. Visit 2	−3.7645	3.1597	0.236
**Treatment**			
Visit 3 vs. Visit 1	13.2248	2.9109	<0.0001
Visit 2 vs. Visit 1	13.2906	2.4401	<0.0001
Visit 3 vs. Visit 2	−0.06575	2.838	0.982

## Data Availability

The data are available in the National Institute for Mental Health National Data Archive. The original contributions presented in the study are included in the article, further inquiries can be directed to the corresponding author.

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
