# Peer review of "Mitigating Weight Gain Side Effects by Reducing Sugar-Sweetened Beverage Consumption in Youth Newly Prescribed Second-Generation Antipsychotic Medication"

_nutrients, 2025, doi:10.3390/nu18010024_

Round 1

Reviewer 1 Report

Comments and Suggestions for Authors

The Introduction would benefit from a clearer contextualization of how antipsychotic-related metabolic changes affect patients beyond weight and metabolic indices. The current framing focuses largely on sugar-sweetened beverage intake and metabolic consequences, whereas a broader perspective on the functional and quality-of-life implications of antipsychotic treatment would help justify the clinical relevance of early metabolic interventions. This is an appropriate place to introduce evidence showing how antipsychotic exposure may influence overall quality of life in psychiatric populations. The systematic review by Sampogna et al.  synthesizes the broader impact of antipsychotic treatment on quality-of-life dimensions, offering an important reminder that weight-related side effects are not isolated phenomena but part of a wider functional burden. 

The Introduction could also reference existing real-world metabolic management efforts in psychiatric settings. This contextualization is useful because it helps the reader understand how the intervention tested here fits among other metabolic strategies already used in routine care. The experience described by Martiadis et al. is appropriate to cite in this section because it reports on a structured outpatient metabolic-management model implemented in everyday clinical practice. Including this perspective would show that lifestyle-based interventions are increasingly considered feasible and relevant in psychiatric care, and that the present study contributes additional controlled data to this area.

In the Methods section certain elements would benefit from clarification. The rationale behind the stratification criteria for randomization should be explicitly stated to help the reader understand why those specific thresholds were chosen. The approach for evaluating antipsychotic adherence requires further detail given the high variability in medication exposure described later in the Results. The handling of missing data related to the pandemic is mentioned, yet the extent to which missingness may have influenced outcomes is not sufficiently discussed.

The Results section is generally clear. The extremely high variability in antipsychotic exposure limits the interpretability of metabolic outcomes and should be highlighted more. Some additional descriptive analyses or sensitivity checks could help quantify how medication switches, discontinuations, or non-adherence influence the results. If not possible, list as a limitation.

The Discussion offers reasonable interpretations, but certain points deserve further development. It should more clearly acknowledge that both groups received healthy lifestyle education and that this may have reduced between-group differences. The improvement in water intake is an important secondary finding and could be expanded upon by discussing biological and behavioral mechanisms that might explain its potential clinical significance. 

The limitations are clearly stated but could mention the likelihood of social desirability bias in self-reported beverage intake and the challenges of implementing lifestyle changes in youth experiencing psychiatric instability.

The Conclusions may benefit from a slightly more cautious tone regarding the intervention’s effectiveness given the variability in antipsychotic adherence and the absence of significant between-group differences in BMI z-scores.

Sampogna G, Di Vincenzo M, Giuliani L, Menculini G, Mancuso E, Arsenio E, Cipolla S, Della Rocca B, Martiadis V, Signorelli MS, Fiorillo A. A Systematic Review on the Effectiveness of Antipsychotic Drugs on the Quality of Life of Patients with Schizophrenia. Brain Sci. 2023 Nov 10;13(11):1577. doi: 10.3390/brainsci13111577. 

Martiadis V, Pessina E, Matera P, Martini A, Raffone F, Monaco F, Vignapiano A, Cattaneo CI. Metabolic Management Model in Psychiatric Outpatients: a Real-World Experience. Psychiatr Danub. 2024 Sep;36(Suppl 2):78-82. PMID: 39378455.

Reviewer 2 Report

Comments and Suggestions for Authors

The paper addresses a topic of great concern for both practitioners and patients when the theme of medication is involved. As already known, weight gain can occur in a myriad of treatments, being pronounced with antipsychotics. After reading the manuscript, I suggest the following:

  • Abstract: the methods section should include more details about the intervention and the sample. It is unclear the duration of the intervention based on the abstract, which could hinder a proper interpretation of results.
  • The introduction needs to be polished. Please, use each paragraph to develop one main idea and conclude it. There are long paragraphs addressing many topics at once, which makes it hard to read. Smooth transitions are needed. Consider subheadings to guide readers.
  • Methods: place participants and design under one heading.
  • There’s very little information about the rationale behind the intervention. Indeed, not a single reference is included. In order to allow for replication, authors could also provide more information as supplementary material.
  • I see as a potential bias the fact that measurements took place when weights in occurred after the pandemic. However, it seems that you did not use this data. I’d rather remove this information or suggest that authors double-check its necessity.
  • Data analysis: report on how the sample size was arrived at.
  • Clearly indicate in tables which statistics are being presented.
  • Although results are described well, the discussion is a bit confusing. Please, rewrite it for clarity and flow. Consider discussing first your primary goals, and only then moving to other topics. I also believe that the attention given to the main results is insufficient and fails to provide enough explanation of the findings for the audience.
  • Finally, check for the journal’s scope and make your paper more aligned with Nutrients. Albeit you only measured a few variables related to nutrition, a more careful dedication into this might enhance is suitability for the chosen publication. Equally, the discussion of alternatives for mental health professionals should be more robust.

Author Response

Review 2

The paper addresses a topic of great concern for both practitioners and patients when the theme of medication is involved. As already known, weight gain can occur in a myriad of treatments, being pronounced with antipsychotics. After reading the manuscript, I suggest the following:

  • Abstract: the methods section should include more details about the intervention and the sample. It is unclear the duration of the intervention based on the abstract, which could hinder a proper interpretation of results.

The abstract has been revised to more clearly delineate the study design and duration of intervention. 

  • The introduction needs to be polished. Please, use each paragraph to develop one main idea and conclude it. There are long paragraphs addressing many topics at once, which makes it hard to read. Smooth transitions are needed. Consider subheadings to guide readers.

Thank you for the feedback.  The introduction has been revised to provide more concise, focused paragraphs. 

  • Methods: place participants and design under one heading.

This change was made. 

  • There’s very little information about the rationale behind the intervention. Indeed, not a single reference is included. In order to allow for replication, authors could also provide more information as supplementary material.
  • We have added references for several aspects of the study design, including the family navigator approach, the water delivery, and modest activity goal setting; along with details of the diet and activity guidelines parents received.   
  • I see as a potential bias the fact that measurements took place when weights in occurred after the pandemic. However, it seems that you did not use this data. I’d rather remove this information or suggest that authors double-check its necessity.
  • We added a sentence in the results to be clearer that we did not use any data obtained post-pandemic, and in the limitation section discussed how this lead to a smaller retention rate than expected due to study discontinuation at pandemic shut down. 
  • Data analysis: report on how the sample size was arrived at.
  • We added the power analysis in the methods section. 
  • Clearly indicate in tables which statistics are being presented.
  • We apologize about that omission and have added the statistics test to the tables. 
  • Although results are described well, the discussion is a bit confusing. Please, rewrite it for clarity and flow. Consider discussing first your primary goals, and only then moving to other topics. I also believe that the attention given to the main results is insufficient and fails to provide enough explanation of the findings for the audience.
  • We have re-organized results and have added a more detailed discussion of the findings for each of the three main outcomes (change in BMI z-score, SSB intake, water intake). 
  • Finally, check for the journal’s scope and make your paper more aligned with Nutrients. Albeit you only measured a few variables related to nutrition, a more careful dedication into this might enhance is suitability for the chosen publication. Equally, the discussion of alternatives for mental health professionals should be more robust.
  • We have added a section to discuss about the impact of hydration on youth outcomes and the impact of APM-induced weight on a broad array of child functioning to better align with the focus of this journal.   

Reviewer 3 Report

Comments and Suggestions for Authors

The study has important scientific value and provides a meaningful contribution to the understanding and prevention of obesity in individuals treated with antipsychotic medications.

However, the paper requires comprehensive revision to improve overall quality.

My suggestions:

1.  In the Introduction, the authors describe results of previous RCTs with aims similar to their own study. However, methodological details appear before the actual background section. The Background should contain only information leading to the research aim, stated clearly in the final sentence.

The Introduction is also long - more emphasis should be placed on the purpose of the study and on the rationale for undertaking this line of research.

2. Figure 1 should be adjusted because some labels appear only partially.

3. The graphs should be prepared in a more aesthetically consistent and clear manner. Please, edit tmem

4.  I suggest also calculating dose equivalents of the antipsychotic medications used and adjusting the results accordingly.

5. There is considerable variability in anripsychotics use during the study period (discontinuities, medication switches), which is a significant confounding factor. This aspect should be more strongly accounted for in the statistical analysis or discussed as a separate limitation.

6, COVID-19 pandemic led to interruptions and differences in the mode of study visits. It may be necessary to discuss how these disruptions affected data completeness and consistency.

7.The results indicate no significant effects of the intervention on BMI or SSB intake. It would be valuable to assess whether the study had adequate statistical power, especially given the high variability in pharmacotherapy.

8. The authors do not indicate whether data on the use of dietary supplements were collected. Supplementation some of nutrients/compounds may influence appetite, dietary behaviors, and overall caloric balance. The absence of such information limits the interpretation of the findings and should be discussed as a potential confounding factor.

9. Given the diversity of drugs and the numerous medication changes during the study, it would be appropriate to account for dose equivalents of antipsychotic medications (e.g., converted to OLA equivalents). This would allow a more precise assessment of the medication burden’s impact on BMI and beverage intake, and help reduce error resulting from treatment heterogeneity.

Author Response

Review 3

  • In the Introduction, the authors describe results of previous RCTs with aims similar to their own study. However, methodological details appear before the actual background section. The Background should contain only information leading to the research aim, stated clearly in the final sentence.
  • The Introduction is also long - more emphasis should be placed on the purpose of the study and on the rationale for undertaking this line of research.

We apologize for including methodological details in the introduction section and have revised this section to better emphasize the rationale and purpose of this study. 

  • 2. Figure 1 should be adjusted because some labels appear only partially.
  • We have corrected this error.
  • The graphs should be prepared in a more aesthetically consistent and clear manner. Please, edit tmem

The figures have been revised so they are more consistent and clear. 

  • I suggest also calculating dose equivalents of the antipsychotic medications used and adjusting the results accordingly.
  • There is considerable variability in anripsychotics use during the study period (discontinuities, medication switches), which is a significant confounding factor. This aspect should be more strongly accounted for in the statistical analysis or discussed as a separate limitation.
  • We have added information in both the discussion and the limitations section about the impact of medication non-adherence and medication discontinuation/switching.
  • 6,
  • COVID-19 pandemic led to interruptions and differences in the mode of study visits. It may be necessary to discuss how these disruptions affected data completeness and consistency.
  • We added to the limitation section that the pandemic lead to lower retention rate since we had to abruptly discontinue study participation for participants enrolled at the time of the pandemic shut down.
  •  
  • The results indicate no significant effects of the intervention on BMI or SSB intake. It would be valuable to assess whether the study had adequate statistical power, especially given the high variability in pharmacotherapy.
  • In addition to the above pandemic challenges, we provided our power analysis to clarify that we did not achieve our initial proposed study sample and how we derived that information.
  • .
  • The authors do not indicate whether data on the use of dietary supplements were collected. Supplementation some of nutrients/compounds may influence appetite, dietary behaviors, and overall caloric balance. The absence of such information limits the interpretation of the findings and should be discussed as a potential confounding factor.
  • Dietary supplements were queried at the home visits. No youth were on supplements, by parent report, during the study.  We have added to the procedure information and the results. 
  • 9. Given the diversity of drugs and the numerous medication changes during the study, it would be appropriate to account for dose equivalents of antipsychotic medications (e.g., converted to OLA equivalents). This would allow a more precise assessment of the medication burden’s impact on BMI and beverage intake, and help reduce error resulting from treatment heterogeneity.

APM-induced weight gain is not clearly dose dependent and significant weight gain is common on low doses.  Also, the prior authorization program restricts initial start and follow up dose according to evidence-based practice (and all youth are enrolled at start of med) so those factors reduce dose ranges. We have also added as a limitation. 

Round 2

Reviewer 1 Report

Comments and Suggestions for Authors

The last version of the manuscript does not show the tracked changes. In this way it is impossible to verify whether the requested changes have been applied. Please provide a new version of the manuscript differentially tracking the changes requested by each reviewer. At this stage I cannot verify the correct modifications stated in the author’s response file. 

Author Response

Comment: The authors have improved the quality of the paper according to the suggestions. However, the main text still contains many typos and inconsistent fonts across different sections. The tables and graphs were not prepared carefully. These issues need to be addressed before publication.

  1. Spelling and grammar were checked with Word spell/grammar check and Grammarly. Typos were found on line 290, and the word their was added before child. On line 425 parent’ was changed to parents’.
  2. The editorial team put the document into the manuscript template, formatted the text (fonts, indents, spacing), and formatted the tables. Although I matched the formatting before copying and pasting the revisions into the template it did not accept the text in the same styles. I was also unable to make changes to the tables formatting since they created the formatting. 

 I reached out to the assistant editor and they were able to correct the formatting of text and tables as reflected in the manuscript uploaded today. 

Email from the assistant editor.

Dear Dr. Bussell,

Thank you for your message. Our editorial team will take care of the formatting of the tables and preparing final layout for the manuscript. Regarding the comment that a brief description of the data is needed in the abstract, we kindly ask you to add it.

Please let us know if you need further clarification.

Kind regards,

Ms. Sadowska Aleksandra

Assistant Editor

E-Mail: aleksandra.sadowska@mdpi.com

Nutrients Editorial Office

Reviewer 3 Report

Comments and Suggestions for Authors

Authors have improved the quality of the paper according to the suggestions. However, the main text still contains many typos and inconsistent fonts across different sections. The tables and graphs were not prepared carefully. These issues need to be addressed before publication

Author Response

Comment: I can see from the responses to reviewers' comments that revisions have been made. I agree with the reviewers to mark the changes in the revision. Please do and send to the reviewers to check. Brief description of data analysis is needed in the abstract as suggested by the reviewers. Please format the manuscript for consistency. Looking forward to the next version of the revision.

  1. The data analysis statement was added to the abstract. Page 1 lines 31-33.
  2. We have attached the first set of responses to the reviewers' comments and added the page and line numbers for easy reference.

Round 3

Reviewer 1 Report

Comments and Suggestions for Authors

thank you for your work

Author Response

There was no comment.